# Research on Ultra-Wideband Radar Echo Signal Processing Method Based on P-Order Extraction and VMD

**DOI:** 10.3390/s22186726

**Published:** 2022-09-06

**Authors:** Qingjie Qi, Youxin Zhao, Liang Zhang, Zhen Yang, Lifeng Sun, Xinlei Jia

**Affiliations:** 1Emergency Science Research Academy, China Coal Research Institute, China Coal Technology & Engineering Group Co., Ltd., Beijing 100070, China; 2Faculty of Electrical and Control Engineering, Liaoning Technical University, Huludao 125105, China

**Keywords:** ultra-wideband radar, return signal, P time strong physical sign information extraction, variational mode decomposition

## Abstract

As a new method to detect vital signs, Ultra-wideband (UWB) radar could continuously monitor human respiratory signs without contact. Aimed at addressing the problem of large interference and weak acquisition signal in radar echo signals from complex scenes, this paper adopts a UWB radar echo signal processing method that combines strong physical sign information extraction at P time and Variational Mode Decomposition (VMD) to carry out theoretical derivation. Using this novel processing scheme, respiration and heartbeat signals can be quickly reconstructed according to the selection of the appropriate intrinsic mode functions (IMFs), and the real-time detection accuracy of human respiratory signs is greatly improved. Based on an experimental platform, the data collected by the UWB radar module were first verified against the measured values obtained at the actual scene. The results of a validation test proved that our UWB radar echo signal processing method effectively eliminated the respiratory clutter signal and realized the accurate measurement of respiratory and heartbeat signals, which would prove the existence of life and further improve the quality of respiration and heartbeat signal and the robustness of detection.

## 1. Introduction

Radar-based life detection technology is a technology that extracts signals related to vital signs (respiration, heartbeat) from radar echoes, which is non-contact, long-distance and capable of penetrating certain media (brick walls, concrete, ruins). The technology has shown broad application prospects in many aspects of military and civilian use [1], such as battlefield search, post-disaster rescue, medical monitoring, anti-terrorism, and stability maintenance, etc. [2,3,4,5]. In recent years, radar as a non-contact vital sign monitoring method has received extensive attention and has been used in various scenarios [6]. At this stage, the radars used for non-contact life signal detection mainly include continuous wave radar and UWB radar [7]. Continuous wave radar has good advantages in signal processing, frequency domain recognition, etc., but it is limited in distance positioning of targets. Ultra-wideband radar is an important remote sensing tool for life detection or non-contact monitoring of life signals [8,9,10]. Compared with continuous wave radar, UWB (ultra-wideband) radar also has great advantages in terms of distance and frequency information The interference capability, resolution and power consumption have been greatly improved [11,12] so that the accuracy and sensitivity of the echo signal are guaranteed. On the other hand, compared with millimeter wave radar and NCS (Nonlinear Chirp Scaling), UWB has a larger detection range and excellent performance [13]. Ultra-wideband radar signals are more sensitive to subtle breathing movements and can also penetrate walls, making UWB suitable for places with many obstacles [14].

When performing vital sign detection, the signal received by the radar not only contains the reflected echo of the target, but also a large amount of complex noise [15,16,17], so certain preprocessing methods are needed to improve the signal-to-noise ratio [18,19]. The vital sign monitoring method of N-order peak capture is also applied to UWB echo signal processing. This method can extract the respiratory frequency and suppress its higher harmonics [20]. Using an improved sensing algorithm for random noise denoising and wavelet packet decomposition, Shikhsarmast et al. [21] proposed a new method to accurately estimate VS parameters under low signal-to-noise ratio conditions such as long-range and through-wall conditions. Xiaolin Liang proposed a method based on ensemble empirical mode decomposition, using FFT (Fast Fourier transform) and Hilbert–Huang to extract human information [22]. Empirical mode decomposition (EMD) can be used to decompose the radar echo signal and filter out the appropriate modal components to reconstruct the breathing and heartbeat signals [23].

In the physical sign extraction method described above, the basis of the extraction is a single slow-time slice containing vital signs or a combination of multiple slices. In the case of complex noise interference, this basis cannot accurately obtain the physical sign information of aperson. In addition, EMD-based decomposition algorithms are prone to spectrum aliasing and end effects, which affect the final decomposition results.

To address the above issues, this paper introduces a UWB radar body sign detection method based on the combination of P sign extraction and VMD. After the radar receives a target echo signal, it de-noises multiple P signals containing vital signs to obtain effective information on the human body; through the use of VMD decomposition, and then according to the center frequency of IMFs (Intrinsic Mode Function), breathing and heartbeat signals are captured through frequency domain peaks.

### Basic Principles of UWB Radar Life Detection

When using UWB radar for life detection in the case of accidents or natural disasters, if a trapped person has signs of life, the network can detect them [24,25,26]. As shown in Figure 1, when the reference distance d0 between the radar system and the human body is constant, movement of the chest cavity can be considered a distance d change, therefore a sign of life. Through continuous measurement and analysis of the change law of d, life can be confirmed, where Δd is the movement distance of the chest cavity caused by breathing and heartbeat.

The chest cavity amplitude change caused by breathing is about 5–15 mm, the chest cavity vibration area is about 50 cm^2^, and its frequency is 0.2–0.5 Hz. The chest cavity amplitude change caused by the heartbeat is about 2–3 mm, the chest cavity vibration area is about 10 cm^2^, and the frequency is 0.8–2.5 Hz [27]. Breathing and heartbeat will cause the chest cavity to produce approximately sinusoidal periodic fluctuations. The distance function is shown below [28]:(1)$$d(t)=d_{0}+\Delta d=d_{0}+A_{r}\sin(2\pi f_{r}t)+A_{h}\sin(2\pi f_{h}t)$$
where $d_{0}$ is the reference distance between the radar antenna and the human thoracic cavity, $A_{r}$ is the respiration amplitude, $A_{h}$ is the heartbeat amplitude, $f_{r}$ is the human respiration frequency, $f_{h}$ is the human heartbeat frequency, and t is the slow time.

The time delay function of transmit and receive is:(2)$$\tau_{v}(t)=\frac{2\times d(t)}{v}=\tau_{0}+\tau_{r}\sin(2\pi f_{r}t)+\tau_{h}\sin(2\pi f_{h}t)$$
where v is the propagation speed of radar waves in the air, and $\tau_{r}=\frac{2A_{r}}{v}$ $\tau_{h}=\frac{2A_{h}}{v}$. Assume that in the measurement area, except for the micro-motion signal of the human body, the surrounding environment is static. The impulse response is the sum of the human target response and the response of the surrounding environment, which can be expressed as:(3)$$h(\tau ,t)=a_{v}\delta (\tau -\tau_{v}(t))+\sum_{i}a_{i}\delta (\tau -\tau_{i})$$
where δ(τ) is the impulse function, $a_{v}$ is the reflection coefficient of the human body, $a_{i}$ is the *i*th environmental factor of the radar wave, $\tau_{v}$ is the time delay corresponding to the human body, and $\tau_{i}$ is the *i*th environmental factor.
(4)$$r(\tau ,t)=s(\tau )*h(\tau ,t)=a_{v}s\delta (\tau -\tau_{v}(t))+\sum_{i}a_{i}s\delta (\tau -\tau_{i})+n(\tau ,t)$$
where s(τ) is the propagation signal of the radar system, ∗ is the convolution operation, and n(τ,t) is the random environmental noise. We can discretely process the radar echo signal to obtain a two-dimensional matrix:(5)$$R(M,N)=r(m\delta_{T},nT_{s})=a_{v}s(m\delta_{T}-\tau_{v}(nT_{s}))+\sum_{i}a_{i}s(\tau -\tau_{i})+n(m\delta_{T},nT_{s})$$
where $T_{s}$ is the sampling interval in the slow time direction, *n* = $t=nT_{s},n=1,2,....,N$, N is the slow time, $\delta_{T}$ is the sampling interval in the fast time direction, and $\tau =m\delta_{T},m=1,2,.....,M$ is the fast time.

## 2. Echo Signal Preprocessing

### 2.1. Pretreatment Method

In addition to the vital sign information, the echo signal received by the UWB radar life detection system also includes interference such as static clutter, huge interference, linear trend, and random noise. In order to effectively remove a variety of clutter, methods such as channel signal subtraction, time average subtraction, linear trend suppression, and signal enhancement are selected. The following are various methods and basic principles.

#### 2.1.1. Channel Signal Subtraction Method

Theoretically, the delay of the background clutter does not change with the scanning time. In order to eliminate the constant component in the scanning process, the range profile subtraction (RPS) method is used to simply eliminate the background clutter of the radar echo signal.
(6)$$Y_{j}^{’}=Y_{j+1}-Y_{j}$$
where $Y_{j},j=1,2,3,....,N$.

#### 2.1.2. Time Average Subtraction

The emitted radar pulse wave will not only reflect on the surface of the human body after touching the human body, but also reflect correspondingly on the stationary parts such as the limbs and trunk of the human body. However, the abovementioned reflected waves are stationary except for the thoracic cavity of the human body, which moves periodically, and the reflected waves hardly change in the fast time axis. Therefore, static background clutter is represented by variables that are uncorrelated with discrete time variables in the slow time dimension.
(7)$$\tilde{S}=\frac{1}{M\times N}\sum_{m=1}^{M}\sum_{n=1}^{N}R[m,n]$$

The result of eliminating static clutter:(8)$$\Omega_{M\times N}=R_{M\times N}-\tilde{S}$$

#### 2.1.3. Linear Trend Suppression

Due to the fluctuation in the amplitude of the radar hardware circuit, this fluctuation will evolve into a signal with a linear trend in the echo matrix. This signal can be divided into different linear functions according to different slopes. After completing the correction of the numerical deviation, the signal still has a linear trend. Aiming at the linear trend problem of the echo signal, the linear trend of the echo matrix is suppressed from the slow time dimension. The corresponding formula is [29]:(9)$$W^{T}=\Omega^{T}-z{\left(z^{T}z\right)}^{-1}z^{T}\Omega^{T}$$
where *z = [z*_1_,*z*_2_*]**, z_1_ = [*1,2,…, *N]^T^**, z_2_ = [*1,1,…, 1*]^T^*.

#### 2.1.4. Signal Enhancement

On the basis of retaining the original frequency characteristics, the low-frequency and high-frequency interference are filtered out, and finally the signal autocorrelation method is adopted to realize the signal enhancement.

The automatic gain control can automatically adjust the gain according to the feedback of the energy value in the signal matrix to strengthen the processing of the vital sign signal, ignore other useless signals, and achieve the purpose of amplifying the energy value of the vital signal.

Human body signs have a certain periodicity, and because the zero-mean noise has no periodicity, it can be reduced to 0 through autocorrelation [15]. The calculation formula of the echo signal after the correlation is:(10)$$R(n^{\prime })=E[x_{m}(n_{1})x_{m}(n_{2})]$$
where 0 ≤ *n*_1_, *n*_2_ ≤ *N* − 1, *n*′ = *n*_1_ – *n*_2_. *E* is the average.

### 2.2. Pretreatment Effect

The grayscale image after preprocessing is shown in Figure 2b. After preprocessing, the background clutter, huge interference, and linear trend are all effectively filtered out, and all kinds of noise interference are effectively suppressed.

## 3. Vital Signs Extraction Algorithm

### 3.1. P Sub-Strong Sign Extraction

Human body sign information is contained in multiple slow-time data slices. Each slice containing sign information can reflect the human body’s respiration and heartbeat frequency information. At the same time, there are certain differences in the sign information of different slices. When slicing data, we increase the error probability of the physical information of the measured human body. Therefore, the method of extracting strong sign information at the time of p-th difference is used to extract the breathing and heartbeat information of the tested person. The flowchart of this method is shown in Figure 3. The signal energy is defined as:
(11)$$e_{n}=\sum_{m=1}^{M}{\left|f_{\left(m,n\right)}\right|}^{2}$$

Among them, $f_{\left(m,n\right)}$ is the amplitude corresponding to the *m*th row and nth column in the matrix, and $e_{n}$ is the energy value corresponding to the *n*th slow-time slice. The energy window with a width of three is used for personnel positioning:(12)$$E_{n}=\sum_{n-1}^{n+1}e_{n}$$

Among them, $E_{n}$ is the signal energy corresponding to the nth sampling window of the echo signal, n=2,3,....,N−1. After determining the maximum energy window position, we use it as the reference position for the first extraction.

When a normal person breathes, exhalation or inhalation will occur every 2 s. Therefore, 2 s is selected as the base time and step length for P extractions. On the slow time slice where the selected reference position is located, the first 2 s of the signal are equally divided to select the starting point of P sign extraction, and the search is performed backwards in steps of 2 s. When searching, we use the reference position and four groups of slow time slices before and after as the search area, select the strongest physical sign information in 2 s in this area, save the sign information and record its position, and use this position as the next search and so on to complete this sign extraction. We then select the next sign extraction starting point until the completion of P extraction.

After a large number of tests in MATLAB, different P values were selected for physical sign information with a total length of 30 s for strong sign extraction, VMD decomposition, and breathing and heartbeat signal extraction. After several experiments and comparisons, it was found that for the same signal, P = 8 is the most suitable value for P strong extraction, which can extract accurate physical sign information in the shortest time. Therefore, in the following experiment, P = 8 is used as the parameter standard for P sub-strong sign extraction.

Let P = 8, eight groups of physical sign information are obtained after strong physical sign extraction, as shown in Figure 4. All the extracted physical sign information has the same changing trend, but there is a certain gap in details. This result has a certain relationship with the distance of the tested volunteer and the preprocessing effect of the echo signal.

### 3.2. VMD Principle

In order to effectively separate the breathing and heartbeat signals contained in the physical signs information, the signal decomposition method is used to decompose them by the method of VMD decomposition. The decomposition process decomposes the signal into several sub-signals with limited bandwidth, which are adaptive, non-recursive The quasi-orthogonal decomposition method is essentially the process of solving variational problems, and the constraint condition is that the sum of IMFs is equal to the input signal f [30]. First of all, assuming that the *K* IMFs obtained by decomposition have a certain center frequency and bandwidth, the center frequency and bandwidth of IMFs are continuously updated during the decomposition process so that the sum of the estimated bandwidth of all IMFs is minimized, and the model can be expressed as [31]:(13)$$\left\{ \begin{array}{l} \underset{\left\{u_{k}\right\}\cdot \left\{\omega_{k}\right\}}{\min}\left\{\sum_{k=1}^{K}{\left|\left|\partial_{t}\left[\left(\partial (t)+\frac{j}{\pi t})*u_{k}(t\right)\right]e^{-i\omega_{k}t}\right|\right|}_{2}^{2}\right\}\\ s.t.\sum_{k=1}^{K}u_{k}(t)=f(t) \end{array}\right.$$

Second, we solve the structural variational problem.
(14)$$\Gamma \left(\left\{u_{k}\right\}\cdot \left\{\omega_{k}\right\}\cdot \lambda \right)=\alpha \sum_{k=1}^{K}{\left|\left|\partial_{t}\left[\left(\partial (t)+\frac{j}{\pi t})*u_{k}(t\right)\right]e^{-i\omega_{k}t}\right|\right|}_{2}^{2}+{\left|\left|f(t)-\sum_{k=1}^{K}u_{k}(t)\right|\right|}_{2}^{2}+\langle \lambda (t),f(t)-\sum_{k=1}^{K}u_{k}(t)\rangle$$

To further solve the variational problem, we iteratively update $u_{k}^{n+1}$, and $\omega_{k}^{n+1}$ $\lambda_{k}^{n+1}$ to seek the “saddle point” of the augmented Lagrangian expression to obtain the optimal solution of the constrained variational model. According to the Parseval/Plancherel Fourier equidistant transform, we convert it to the frequency domain and solve the minimum problem:(15)$${\hat{u}}_{k}^{n+1}(\omega )=\frac{\hat{f}(\omega )-\sum_{i\neq k}\hat{u}(\omega )+\frac{\lambda (\omega )}{2}}{1+2\alpha {\left(\omega -\omega_{k}\right)}^{2}}$$

Using the same principle to solve the problem of the minimum value of $\omega_{k}^{n+1}$ 1, the center frequency is obtained as:(16)$$\omega_{k}^{n+1}=\frac{{\int_{0}^{\infty }\omega \left|{\hat{u}}_{k}(\omega )\right|}^{2}d\omega }{{\int_{0}^{\infty }\left|{\hat{u}}_{k}(\omega )\right|}^{2}d\omega }$$

Finally, the inverse Fourier transform is performed on formula (14) to obtain $\left\{u_{k}(t)\right\}=\left\{u_{1}(t),u_{2}(t),\ldots ,u_{K}(t)\right\}$; that is, the *K* IMFs components to be sought, $\left\{\omega_{k}\right\}=\left\{\omega_{1},\omega_{2},\ldots ,\omega_{K}\right\}$ is the center frequency corresponding to each IMF component.

For VMD decomposition, the number of IMFs *K* and the penalty factor α are the key parameters, and different parameter combinations will affect the decomposition results. If the number of decomposition modes *K* is too small, spectrum aliasing will occur; if the value of *K* is too large, it will cause over-decomposition and produce some useless false components. At the same time, the set penalty factor α value will also affect the bandwidth of IMFs components. If the value of α is not selected properly, the bandwidth of each eigenmode component obtained after decomposition will be unstable, Reasonable setting of two key parameters is the key to the successful implementation of the VMD method.

After a large number of experimental verifications, in general, when *K* = 5 and α = 20,000 [32], the confidence of the physical signs can be better decomposed. In the results obtained by applying this parameter combination, there will be a center frequency in the human breathing. If there are no IMFs in the range of the heartbeat frequency band, we increase the value of *K* until IMFs that reflect the body’s physical information are obtained.

Take *K* = 5 and *α* = 20,000. We decompose the physical sign information; the result is shown in Figure 5, and the frequency characteristics of IMFs are shown in Figure 6. Combining Figure 5 and Figure 6, it can be seen that after VMD decomposition, the original physical sign information is decomposed into five IMFs according to frequency characteristics. Each component has its own center frequency, and the spectrum aliasing phenomenon is relatively weak.

### 3.3. Reconstruction of Breathing and Heartbeat Signals

When VMD is decomposed, the center frequency of each IMF will be updated and finally stabilized at a value. The range of human breathing is 0.2–0.5 Hz, and the range of a heartbeat is 0.8–2.5 Hz. In the obtained IMFs, according to the changes of their respective center frequencies, IMFs whose center frequencies occupy more than 80% of the respiratory and heartbeat frequency bands are selected for reconstruction, and the final respiratory and heartbeat signals are obtained. According to the above method, VMD decomposition and reconstruction are performed on the obtained eight sets of physical signs, respectively, and the obtained breathing signal is shown in Figure 7, and the heartbeat signal is shown in Figure 8.

Combining Figure 7 and Figure 8, it can be seen that the eight sets of respiratory signal traces are completely consistent with very few differences, while the overall trend of the eight sets of heartbeat signals is the same, although there are some subtle differences. The reason is that compared with the respiratory signal, the heartbeat signal is more difficult to extract.

When extracting the frequency of breathing and heartbeat, the reconstructed breathing and heartbeat signals are respectively subjected to FFT transformation, and the resulting spectrograms are shown in Figure 9 and Figure 10. In the breathing range, the peak value of the spectrum of the breathing signal is obtained to determine a plurality of corresponding frequencies and amplitudes, and finally the frequency with the most peaks is determined as the breathing frequency. If the peaks appearing at multiple frequencies are all the maximum times and the one with the largest average amplitude is selected as the final frequency andthe average amplitude is still the same, the corresponding frequencies are averaged. According to the above method, the breathing rate of the trapped person can be determined, and the search range can be changed to the frequency range of the heartbeat to complete the extraction of the heartbeat frequency.

It can be seen from Figure 9 and Figure 10 that within the frequency band of the respiratory signal, only eight peaks are obtained at frequencies of 0.299 Hz and 0.4332 Hz, and no peaks appear at other frequencies. However, the average amplitudes corresponding to the two frequencies are 12.205 and 1.798, respectively, so 0.299 Hz is selected as the respiratory frequency. The heartbeat signal’s frequency is rather messy, with 15 peaks appearing in many locations. The average amplitude at the frequency of 1.233 Hz is 0.670, which is higher than the amplitude at other frequencies, and is selected as the heartbeat signal frequency.

The general flowchart of the vital signal extraction algorithm shown above is shown in Figure 11.

## 4. Experimental Verification

### 4.1. Experimental Equipment

This experiment is composed of an ultra-wideband radar system built with the NVA-6100 chip and a fast time equivalent conversion distance of 0.4 cm. Both the signal transmitting part and the receiving part of the system adopt Vivaldi antennas. The specific NVA-6100 parameter information is shown in Table 1.

### 4.2. Experimental Test Platform

In order to verify the accuracy of the algorithm, the experimental platform built is shown in Figure 12. The height of the model is 1.3 m. The chest cavity contains a self-made simulated lung controlled by a single-chip microcomputer code. The code controls the vacuum pump to inflate and deflate regularly at a certain frequency to simulate the human breathing signal.

The generator of the life micro-motion breathing signal includes a micro-vacuum pump and a balloon. The Y-shaped tube and the balloon are connected by a catheter to simulate the lungs for ventilation. The air pump is used for inhalation and exhalation, respectively. The expiratory frequency directly converts the life frequency signal generated by the air pump motor simulation into a visible expansion and contraction signal, which truly simulates the human breathing process.

The microcontroller uses the STM32F10X series to generate a PWM wave to control the shut-off state of the input voltage of the air pump. The currently set PWM wave frequency is 1/3 Hz; that is, a breath is 3 s in total, and the corresponding air pump is inflated for 1.5 s and deflated for 1.5 s. The cycle or the inflation and deflation time is adjustable in the program.

During the experiment, the UWB radar system was placed on a table 0.85 m from the ground. The model was facing the radar system, and the respiratory frequency of the human simulated lungs was set to 0.3 Hz by controlling the single-chip microcomputer. By controlling the motor of the track, the human body model is continuously moved to different positions on the slide rail, and the position is marked. Finally, the radar system is used for related tests. The distance test results of the human body model of the experimental platform are shown in Table 2:

According to the data analysis in Table 1, as the setting distance increases, the measurement error gradually becomes larger. The relative error of the average distance measured by this experimental platform is 1.17%. The corresponding test results are shown in Table 3.

It can be seen from the data in Table 2 that this algorithm can separate the simulated breathing frequency from the echo signal, and its average relative error is 4.47%. The reason for the error is the lack of stability of the simulated lung, the low accuracy of simulation, the small vibration area of the simulated chest cavity connected to it, and the thin vibration plane, which makes the information of the radar echo signal weak.

### 4.3. Practical Test

In order to verify the feasibility and accuracy of this algorithm for actual scenes, an experimental site as shown in Figure 13 was set up.

The volunteer who participated in this measurement was a man weighing 60 kg, height 170 cm, and age 27 years old. After several measurements, the average value was taken to obtain a breathing rate of 0.2667 Hz (16 beats/min) in a calm state and a heartbeat. The frequency is 1.133 Hz (68 times per minute). During the experiment, the tested volunteers were sitting in front of the radar system with their chest cavity facing the transceiver antenna of the radar system, staying still and breathing evenly, while minimizing body movement as much as possible. According to the above measurement method, the volunteers are located at the 1 m, 2 m, 3 m, 4 m, and 5 m marks in the test on the experimental platform, and the obtained echo signals are processed, respectively, The actual distance and frequency measurement results are shown in Table 4. The respiration and heartbeat spectra at various distances are shown in Figure 14:

It can be seen from Table 3 that as the distance increases, the actual measurement error gradually increases, and the average relative error is 1.52%. Combining the measured data in Table 2, it can be seen that the data are all greater than the actual distance, while the measured data in Table 4 are all less than the actual distance. The reason is that during the experiment, the distance is set based on the center position of the model and the volunteer, and the vibration position of the model is the artificial lung inside it, not the frontmost chest position, which is slightly farther than the center position. The voluntary vibration position is the chest cavity, which is closer to the radar system than the reference point. At the same time, the volunteers will inevitably move slightly during the measurement, which will also cause certain errors in the measurement results.

It can be seen from Table 5 that the average relative error of the actual measured respiratory frequency is 2.92%, and the average relative error of the heartbeat frequency is 4.24%. As the measurement distance increases, the error gradually increases. It can be seen from the respiration spectrum shown in Figure 14 that as the measurement distance continues to increase, the spectral similarity of the respiration signal becomes lower and lower, but there are still significant peaks. For the frequency spectrum of the heartbeat signal, as the distance increases, it gradually becomes messy, and the similarity will also decrease. However, according to the peak law of the frequency spectrum, the corresponding heartbeat frequency can still be observed.

## 5. Conclusions

Radar-based detection technology is an effective way to monitor vital signs in different scenes. In this paper, a UWB radar echo signal processing algorithm integrating P sub-strong sign extraction with VMD is proposed, which can extract the strongest sign information, and accurately estimate respiration according to the reconstructed respiration and heartbeat signals of the P group and heartbeat frequency. Analysis of the results leads to the following conclusions:(1)Through many tests of MATLAB software, P = 8 is a more suitable parameter for the algorithm. The larger the P value, the higher the accuracy of breathing and heartbeat.(2)An experimental platform was built to simulate a human breathing signal of a certain frequency with an artificial lung. The average relative error of the distance measured by the algorithm is 1.17%, and the average relative error of the measured respiratory frequency is 4.47%.(3)In the actual test with a volunteer under different distance states, the average relative error of the distance of the algorithm is 1.52%, the relative error of the measured respiratory frequency is 2.92%, and the average relative error of the heartbeat frequency is 4.24%.

For the signal processing algorithm, we plan to continue to optimize the details of the P extraction and add more adjustable parameters to make it have a wider range of application scenarios. We also plan to optimize the selection of the main parameters of the VMD algorithm, quantify the decomposition results, and make the parameter selection more reasonable.

## Figures and Tables

**Figure 1 sensors-22-06726-f001:**
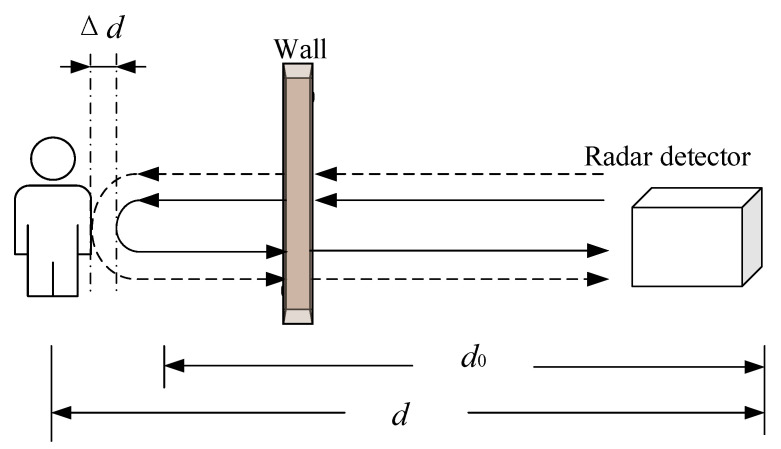
Detection principle.

**Figure 2 sensors-22-06726-f002:**
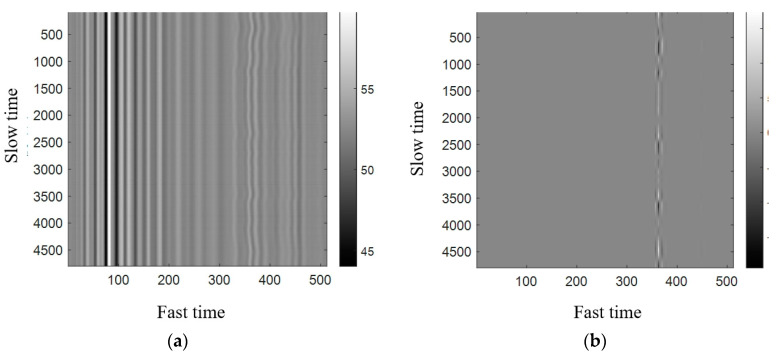
Comparison of grayscale maps before and after preprocessing. (**a**) Grayscale map before preprocessing. (**b**) Grayscale map after preprocessing.

**Figure 3 sensors-22-06726-f003:**
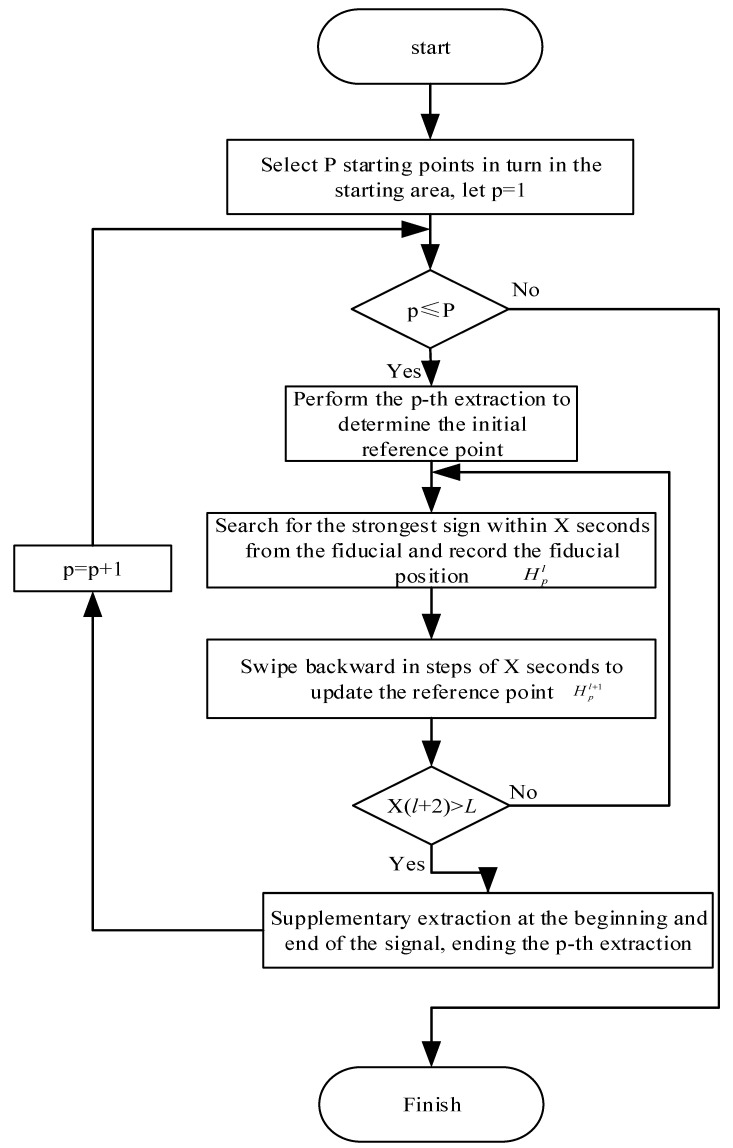
Signal acquisition process.

**Figure 4 sensors-22-06726-f004:**
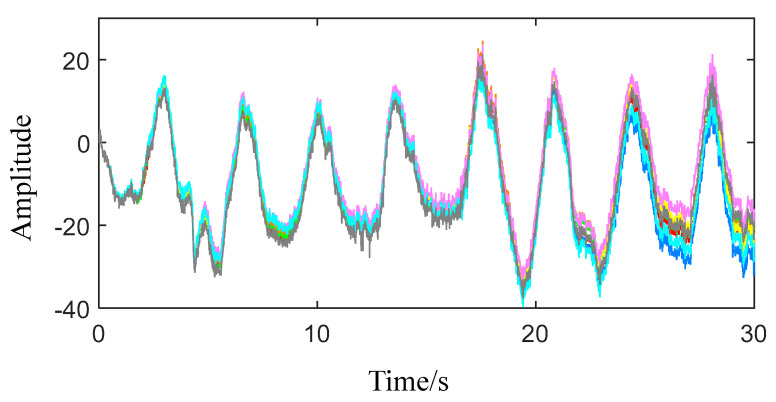
Results of 8 times strong physical sign extraction.

**Figure 5 sensors-22-06726-f005:**
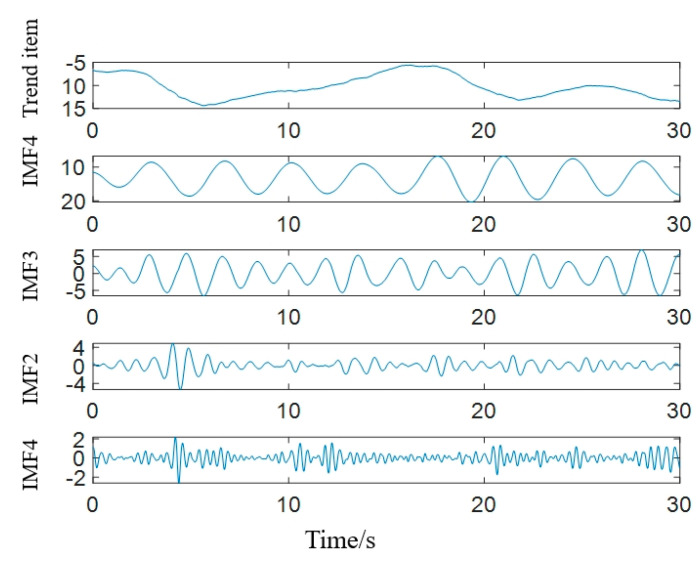
Results of VMD decomposition.

**Figure 6 sensors-22-06726-f006:**
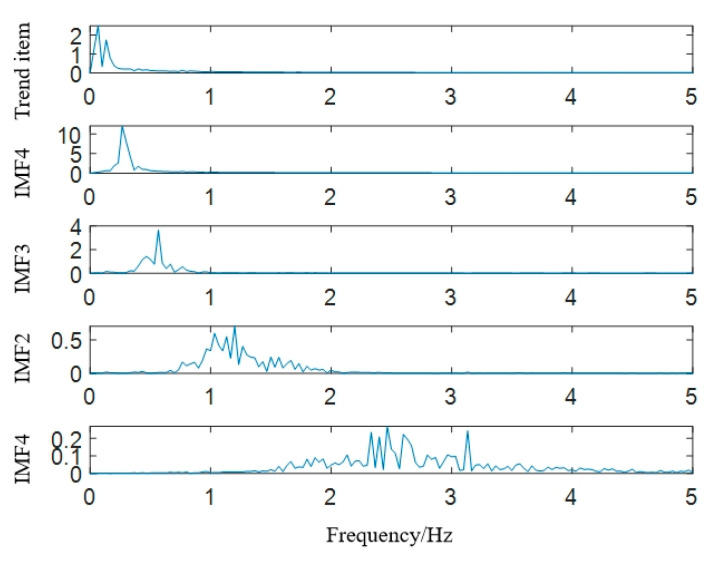
Spectrum of IMFs.

**Figure 7 sensors-22-06726-f007:**
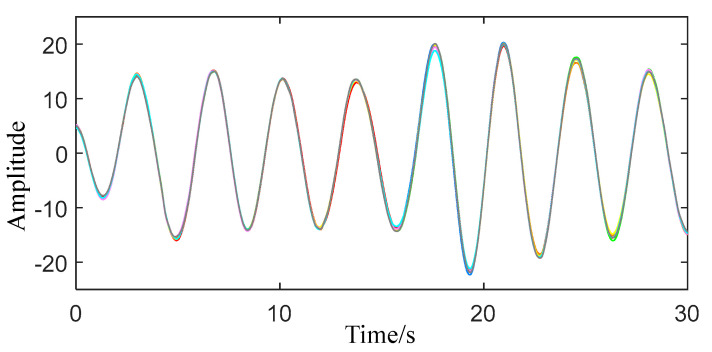
Reconstructed respiratory signal.

**Figure 8 sensors-22-06726-f008:**
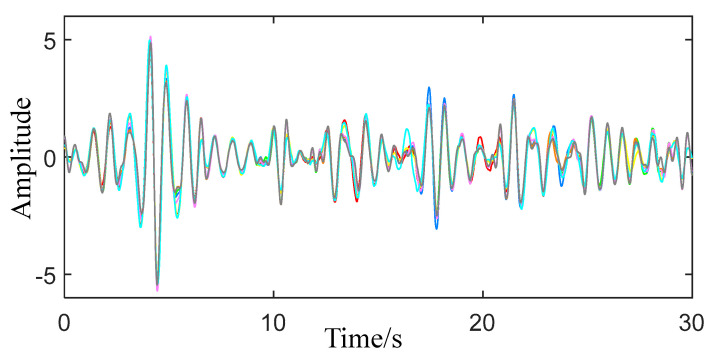
Reconstructed heartbeat signal.

**Figure 9 sensors-22-06726-f009:**
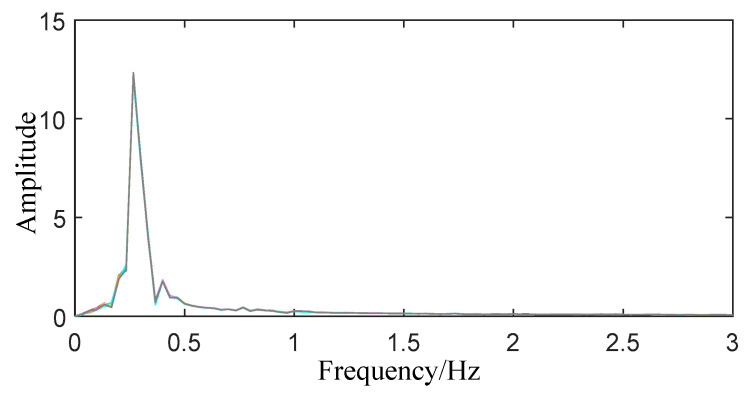
Spectrogram of respiratory signal.

**Figure 10 sensors-22-06726-f010:**
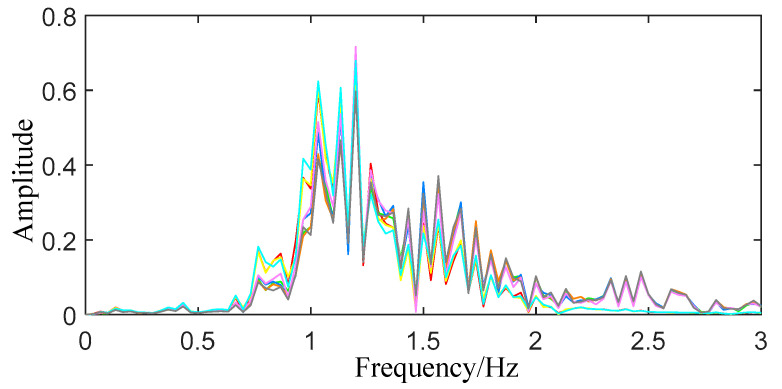
Spectrogram of heartbeat signal.

**Figure 11 sensors-22-06726-f011:**
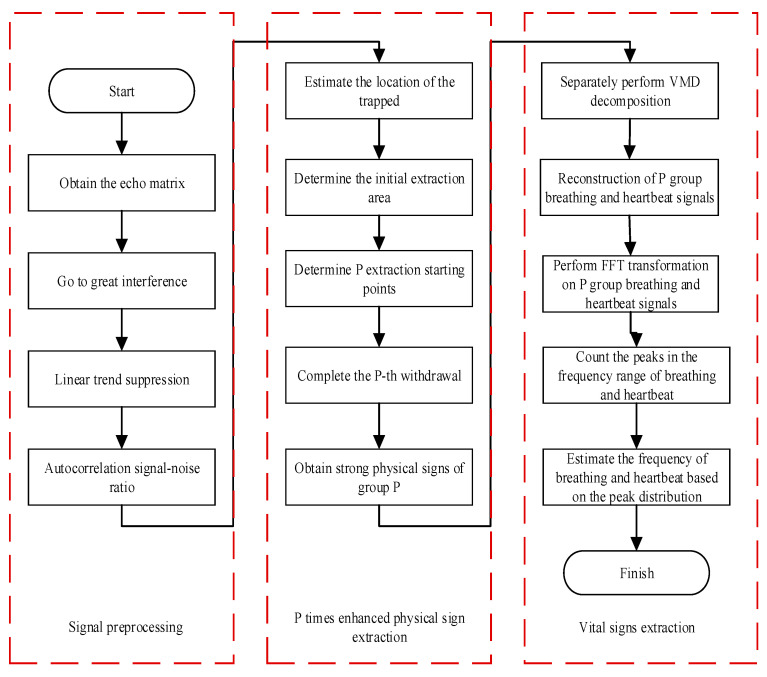
Flowchart of vital signal extraction algorithm.

**Figure 12 sensors-22-06726-f012:**
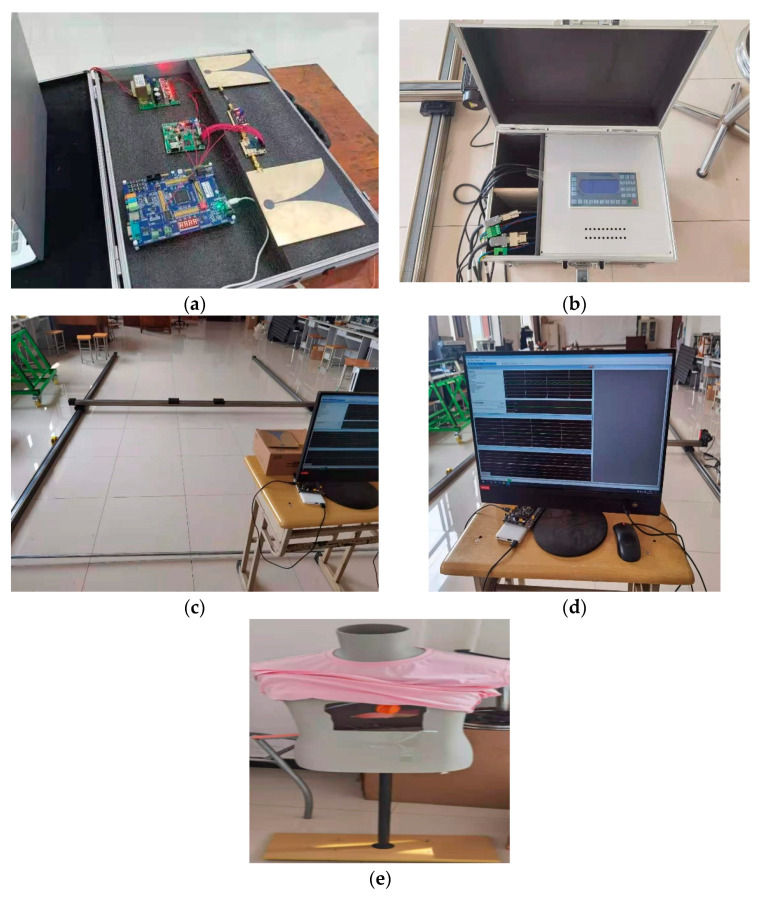
Experiment platform. (**a**) Radar transceiver equipment. (**b**) Experimental platform control cabinet. (**c**) Position mobile module. (**d**) Host computer. (**e**) Breathing and Heartbeat Simulator.

**Figure 13 sensors-22-06726-f013:**
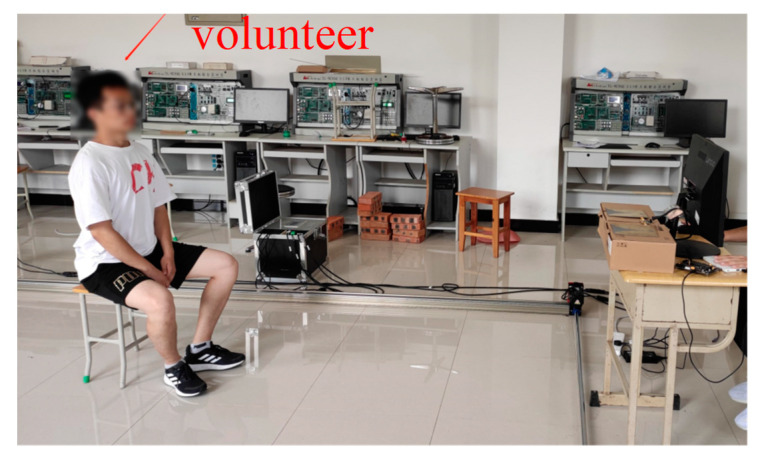
Experimental scene.

**Figure 14 sensors-22-06726-f014:**
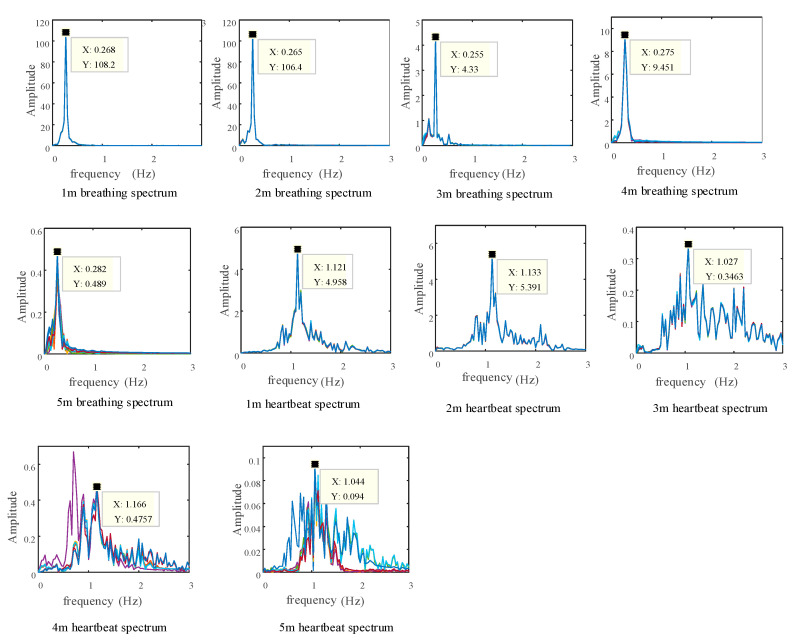
Actual measured spectrum.

**Table 1 sensors-22-06726-t001:** Parameters of UWB radar system.

Parameter	Numerical Value
−10 dB cutoff frequency	0.45 Ghz
−10 dB upper cutoff frequency	3.55 Ghz
Output Power	−14 dBm
Fast time domain sampling frequency	20 Hz
Slow time domain sampling frequency	152.6 Hz
Fast time domain sample length	512
Pulse repetition frequency	48 MHz
Equivalent Distance Resolution	4 mm
Vivaldi antenna gain	6 dBi

**Table 2 sensors-22-06726-t002:** Experimental platform distance test results.

Group	Distance (m)	Measured Distance (m)	Distance Error (m)	Relative Error
A	1	1.016	0.016	1.60%
B	2	2.024	0.024	1.20%
C	3	3.028	0.028	0.93%
D	4	4.044	0.044	1.10%
E	5	5.052	0.052	1.04%

**Table 3 sensors-22-06726-t003:** Experimental platform frequency test results.

Group	Measured Frequency (Hz)	Error (Hz)	Relative Error
A	0.292	0.008	2.67%
B	0.316	0.016	5.33%
C	0.290	0.010	3.33%
D	0.283	0.017	3.67%
E	0.278	0.022	7.33%

**Table 4 sensors-22-06726-t004:** Actual scene distance test result.

Group	Distance (m)	Measured Distance (m)	Distance Error (m)	Relative Error
A	1.0	0.980	0.020	2.00%
B	2.0	1.968	0.032	1.60%
C	3.0	2.960	0.040	1.33%
D	4.0	3.944	0.056	1.40%
E	5.0	4.936	0.064	1.28%

**Table 5 sensors-22-06726-t005:** Actual scene frequency test results.

Group	Respiratory Rate (Hz)	Breathing Error (Hz)	Relative Breathing Error	Heartbeat Frequency (Hz)	Heartbeat Error (Hz)	Relative Error of Heartbeat
A	0.268	0.001	0.37%	1.121	0.012	1.06%
B	0.265	0.003	1.12%	1.133	0.000	0.00%
C	0.255	0.012	4.49%	1.027	0.106	9.36%
D	0.275	0.008	3.00%	1.166	0.033	2.90%
E	0.282	0.015	5.62%	1.044	0.089	7.86%

## Data Availability

The test data used to support the findings of this study are available from the corresponding author upon request.

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
