# Peer review of "Research on Ultra-Wideband Radar Echo Signal Processing Method Based on P-Order Extraction and VMD"

_sensors, 2022, doi:10.3390/s22186726_

Round 1
Reviewer 1 Report
An UWB radar echo signal processing algorithm integrating P sub-strong sign extraction with VMD is proposed, which can extract sign information, accurately estimate the respiration according to the reconstructed respiration and heartbeat signals. In general, the novelty of the manuscript needs to be further enhanced. Also there are some issues should be improved in the aspects of the description and grammar. Some specific suggestions for improvement are listed as follows:
1. In the part 2.1.2, time average subtraction formula is incorrect. It doesn't make sense to subtract a value from the entire two-dimensional array, and the subtract average alone with sliding window should be used in the slow time dimension in this step.
2. The authors claim that, for the same signal, P=8 is the most suitable value for P strong extraction, which can extract accurate physical sign information in the shortest time. Please provide the detailed explaination in the manuscript.
3. ‘K’ and ‘a’ in this paper are determined based on experience, but respiratory and heartbeat signals are vary from different person and different instant. Is this scientific each decomposition parameter is all determined by k=5, and a=20000? And, when the ‘a’ value is large enough to lead the bandwidth of IMF component is too narrow, will some subtle details of vital sign be lost?
4. Author need provide the picture of the self-made simulated lung.
5. Several of the figures are of poor quality and authors should provide more clear versions.
6. Please carefully proofread the manuscript to eliminate typos and grammar mistakes. It is suggested to invite native speakers of English to polish the manuscript to make it easier to read and understand.
7. Extend the chapter of CONCLUSION to discuss the shortcomings of the proposed method and the future work.
8. In the Figure 6 and Figure 7, there are 8 sets of respiratory and heartbeat signals. The authors need to describe how these 8 groups of signals obtained.
9. Some references are not cited in the text, such as reference 8.9.10.
Reviewer 2 Report
(1)In the part 2 Echo signal preprocessing,the author introduces many signal preprocessing methods, which have been published in many literatures. There is no need to discuss them.
(2)In the part 3.1. P sub-strong sign extraction, the proposed method is introduced, but it is too simple and unclear. It is suggested that the author discuss the cover part in detail.
(3)The VMD method is relatively mature, and the article has no substantive innovation.
(4)In the experimental part, the signal obtained by human respiration is not a single frequency signal, and has harmonic components. How to prove that the detected heartbeat signal is not the harmonic component of respiratory signal?
Reviewer 3 Report
The paper presents a method for the extraction of vital signs from a UWB radar.
I have some comments that should be improved:
1. Some acronyms should be defined the first time that appears for example VMF and IMF in the text. These acronyms are defined in the abstract but not in the text.
2. In the abstract and in the title, the concept of P time is a bit confounding. It must be explained better in the abstract.
3. The extraction of P slices can be explained better with some schema or diagram. This is the main novelty of the work for example compared with [32].
4. Some small mistakes should be corrected:
In (6), remove the last “N”.
Several phases must be finished with an endpoint (eg. In line 220).
In line 240, insert a space in “to obtain”
5. Some details and configuration of the NVA-6100 radar should be included in section 4.1.
6. The chest simulator should be shown in Fig.11. Only the control unit is shown in this figure.
7. How is it measured the heartbeat used as a reference in the comparison in the tables in the experiments?. The instrument or method used should be indicated in the text.
8. Some results as a function of body orientation can be included. The heartbeat signal is difficult to obtain with poor SNR signals, for example when increases the distance or when the radar is not oriented to the chest.
9. Some comments about the influence of the movement in the proposed method should be included.
10. Usually, harmonics of breathing signals and other intermodulation products can fall in the heartbeat spectrum frequency range. All the results shown in fig.13 present a similar breathing rate (0.26-0.28 Hz). What happens if increases the breathing rate? What is the robustness in front of the harmonic interference?
11. It is possible to detect apneas and the heart rate during apnea?
Round 2
Reviewer 1 Report
The authors responded to the most comments of reviewer, but there is still space for improvement. Specific suggestions are provided below.
1. MDPI's English Editing Services are recommended to polish the language.
2. Still some figures' quality need to improve, such as Fig. 5, Fig. 6.
3. The authors need to explain how to generate simulated person's respiration and heartbeat signals simultaneously with an air pump and a balloon in the next version.
4. In the line 12 of page 1, 'a' should be replaced by 'an'.
5. In the line 21 of page 1, '.' should be replaced by ','.
6. In the line 45 of page 1, full spelling of 'NCS' need to be provided when it firstly appear in the manuscript.
7. In the line 4 Table1, page 12, there should be 'Slow time domain sampling frequency'.
8. In the line 358-359 of page 14, there is a mistake of broken sentences need to be corrected.
Reviewer 3 Report
The authors have responded to my comments, so I have no additional comments.
Author Response
Thank you for your valuable comments and recognition of this paper.